# Image Abstraction through Overlapping Region Growth

Rosa Azami*        David Mould†

Carleton University, Ottawa, Canada

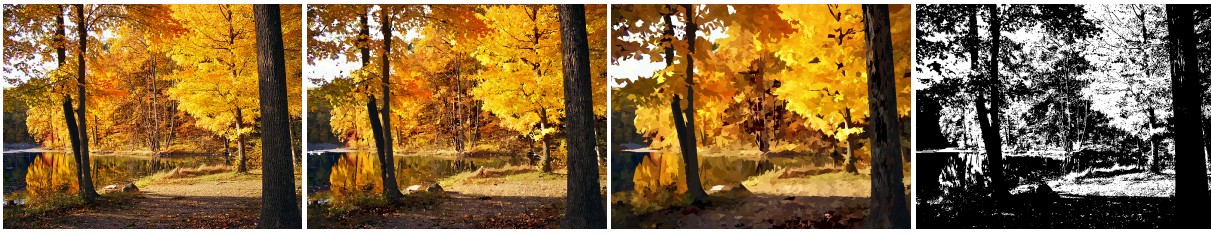

Figure 1: Abstraction result. Left to right: input image, modal region colors, simplified boundaries, black and white.

## ABSTRACT

We propose a region-based abstraction of a photograph, where the image plane is covered by overlapping irregularly shaped regions that approximate the image content. We segment regions using a novel region growth algorithm intended to produce highly irregular regions that still respect image edges, different from conventional segmentation methods that encourage compact regions.

The final result has reduced detail, befitting abstraction, but still contains some small structures such as highlights; thin features and crooked boundaries are retained, while interior details are softened, yielding a painting-like abstraction effect.

**Keywords:** Non-photorealistic rendering, Image stylization, Segmentation, Abstraction.

**Index Terms:** I.3.3 [Picture/Image Generation]—; I.4.6 [Segmentation]

## 1 INTRODUCTION

Image stylization has seen a tension between abstraction, which removes detail from input images, and media emulation, which often adds detail such as watercolor texture or brushstroke direction. However, while the resulting images can contain fine-scale details such as paint texture, such details are often dissociated from any fine-scale detail that had been present in the input. The resulting images often show uniformity in shape and size of the abstraction primitives. This issue appears both in filter-based and region-based methods. Abstractions by filter-based methods may blur edges. Region-based abstractions often generate uniform-size regions which cannot capture small-scale structures and textures. We present an automatic oversegmentation algorithm that can create regions with highly irregular shapes and sizes; the resulting regions can then be used as stylization primitives. Sample results from our approach appear in Figure 1.

Our approach uses region-based abstraction to modify an input photograph, where the image plane is covered by overlapping irregularly shaped regions that approximate the image content. Unlike traditional segmentation methods, or superpixel methods such as

---

*e-mail: rosa.azami@carleton.ca
†e-mail: mould@scs.carleton.ca

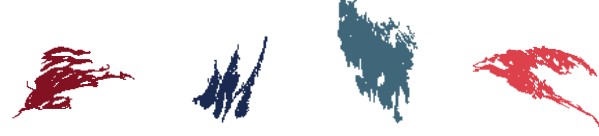

Figure 2: Irregular regions generated by our method.

SLIC [1], we deliberately create non-compact regions, with the jaggedness and complexity of region shapes creating significant visual interest. Figure 2 shows some sample region shapes generated with our method. Image detail is preserved because the region boundaries conform to image edges, while variation in final region size allows very small details to be kept. We do not add any detail based on simulated media, though that could certainly be added if desired. Our regions have the following properties:

- Irregularity: generated regions have high irregularity in shape and size.

- Detail preserving: capture the small details of the image, like textures or thin features.

- Structure preserving: can conform to edges within the image, which enables capturing the structures.

Our process is entirely automatic. From the initial image, we compute a detail map by subtracting a smoothed version. Connected components of a thresholded detail map generate seeds for a region growth process. Seeds are ordered based on the residual values, and seeds with smaller residual values are expanded first. When all regions have been found, we assign a color to the visible portion of that region.

This paper makes the following contributions:

- Overall, we propose a mechanism for abstracting images through highly irregular region shapes that nonetheless conform to image content. Our approach can be seen as a hybrid of stroke-based and region-based stylization methods.

- We introduce a region-growing algorithm that generates intricate region shapes that conform to local detail. Unlike traditional oversegmentation methods, we emphasize the irregularity of region shape even in uniform areas.

- We perform a preliminary investigation of stylizations that involve manipulation of the regions' colors or boundary shapes. While this approach is in its infancy, the early results appear promising.

The remainder of the paper is organized as follows. In Section 2, we discuss related work. We then describe our method in Section 3 and present results from our abstraction process in Section 4, followed by discussions of the results in Section 5. We close in Section 6 and suggest directions for future work.

## 2 PREVIOUS WORK

In the last two decades, there has been considerable progress in the science of generating stylized images. The immensity of the literature makes it difficult to discuss all related research in this section. However, we attempt to cover the work that is most related to ours in both results and approach.

**Stroke-based methods** A common methodology involves repeatedly placing brush strokes with different color, size, and orientation [4, 12, 13, 20, 33, 34]. Various artistic styles can be simulated in this way. Haeberli [12] placed brush strokes in a stochastic distribution in the neighborhood of a manually-driven cursor. Hertzmann [13] placed curved strokes following image content, rendering smaller brush strokes over larger ones. Zheng et al. [34] applied strokes in the context of 3D painting and animation, placing long, variable-sized brush strokes and manipulating stroke characteristics according to movements of the camera.

**Region-based methods** Many region-based methods exist. In seminal work in this area, DeCarlo and Santella [7] performed segmentation [2] at different scales and used an overlap rule to construct a hierarchy. Wen et al. [30] applied an interactive segmentation, where the user specifies which regions are background/foreground. Arty shapes [28] used simple shapes fitted to regions. Faraj et al. [10] generate stylized images from simplified shapes organized in a tree structure. They use a dictionary to extract shapes and simplify the abstraction by removing small shapes based on a threshold value.

**Filter-Based methods** Several filter-based methods have been proposed for abstract and artistic images [6, 14, 17, 22, 24, 25, 27, 31]. Papari et al. [25] introduce a painterly filter based on the Kuwahara filter [16]. They employed different weighting functions to preserve corners and edges. Later, Kyprianidis et al. [17] present a generalization of the Kuwahara filter that preserves the local structure and directional image features, providing better content preservation. Winnemöller et al. [31] present an automatic, real-time video and image abstraction, by approximation to anisotropic diffusion. Orzan et al. [24] generated abstracted images by performing a Poisson reconstruction on extracted edges of the image. Their technique also produces different stylizations such as drawing and watercolor style. Semmo et al. [27] proposed a method for filtering in an oil painting style. They quantize the input image color, then add a paint texture derived from local image orientations.

*Smoothed local histogram filters*, introduced by Kass and Solomon [14], perform filtering based on the content of local image histograms. Histogram modes can be useful for stylization, segmentation, and noise reduction purposes.

The geodesic filter [6] and its variant, *cumulative range geodesic filter* [22], both smooth the image while keeping edges sharp. Textured regions are abstracted through CRGF, but the irregular shapes and ragged edges are kept. Since we are interested in creating irregular region shapes, we will use the CRGF filter as a basis for region growth.

**Reduced color palettes** Black and white [19, 23, 32] images are simplified into only two colors. Rosin and Lai [26] proposed an algorithm to add spot colour to grayscale or monochromatic images. Further techniques for reduced-palette stylization are described in the survey by Lai and Rosin [18].

Capturing image texture was an initial objective for our region generation process. Some learning based methods [9] generate stylized images by transferring textures from one image to another. In future, we hope to investigate a hybrid of learning-based and region-based techniques.

## 3 PROPOSED METHOD

Our method is a region growth approach guided by local colors, where regions are the key elements of the abstraction. Given an input image $I$, we want to generate a set of regions $\{R_i\}$ that capture its important structures. Image textures are presented by the irregular shape of the regions. Our method has three main steps:

- *Seed placement*: we identify a starting point for each region; this is done by finding connected components from a thresholded detail layer. Each connected component above a minimum size generates one seed.

- *Growth mechanism*: a region grows until a stopping criterion is reached. We halt expansion locally when the incremental cost becomes too large; when expansion is not possible in any direction, the region is finished.

- *Rendering*: we place regions in order of size: on the bottom, the largest regions cover the background, and smaller ones come on top providing detail. The final color of each region is the modal color of its visible pixels.

Figure 3 provides a visual depiction of these stages. In the following, we describe each step of the algorithm in more detail.

### 3.1 Seed Placement

In an effort to preserve image details, we used the image residuals to guide seed placement. We blur the input image and subtract the original, producing a detail layer. The detail layer is separated into negative and positive portions and thresholded, leaving us with three possible labels for every pixel: high positive value; high negative value; and neutral value.

We then find all connected components with positive and negative labels, discarding components that are too small (less than 10 pixels in size). We compute one region per connected component, starting the region growth at the component's median pixel position.

### 3.2 Growth Mechanism

The seeds are now treated individually, with a region grown around each one, following a best-first order in an 8-connected graph. We continue adding pixels to the region, ordered by increasing cumulative range geodesic distance [22], until growth in every direction has halted. Effectively, this process involves computing shortest paths where the edge weight to a node $p$ is the color distance to the seed, $\|I(p) - I(s)\|$. In our process, we used the squared color distance in Lab space; we found that squaring the distance led to greater ability to distinguish fine detail.

Each seed is assigned a unique region ID, used to label each pixel in its region. A given pixel can belong to many regions simultaneously. In the end, we assign each pixel the ID of its smallest region.

#### Region growth termination

Our region growth process can be viewed as a single-source shortest path problem, where we compute range geodesic distances to a tree of nodes surrounding the seed. We expand the region by recursively adding the neighbours of each node, tracking the total path cost of each node from the seed. When the incremental cost along the growth path exceeds a per-region parameter $T$, we halt further growth along that path. When all growth paths have been halted, the region is considered complete. In our implementation, we do

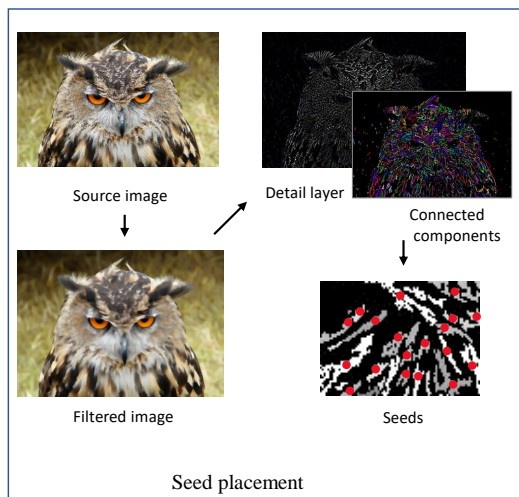

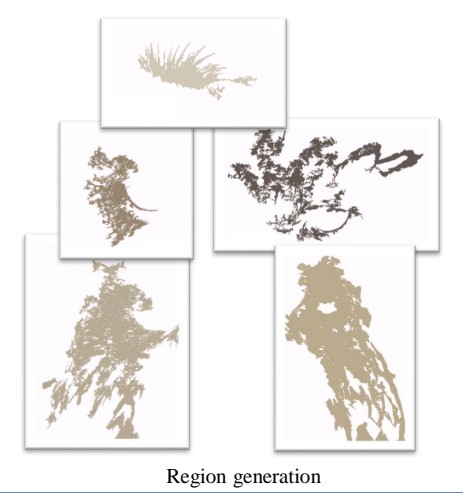

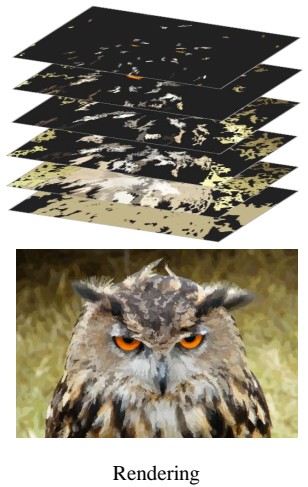

| Seed placement | Region generation | Rendering |

Figure 3: Region-based abstraction pipeline. We compute a detail layer from input image and its filtered one. Then, we generate initial seeds for each region from the connected components of this detail map. For each seed, we expand a region. When all regions have been found, we color the output by assigning to each region the mode of the pixel values in the visible portion of that region.

not explicitly terminate growth through a node; rather, the boundary nodes that we are considering adding are stored in a priority queue, and we implicitly halt progression by not adding potential successor nodes into the queue.

We decide whether or not to locally halt progression based on whether or not the cost of going in the current direction is considered to be large in the context of the current region. We operationalize "large" as follows. The first time a node added to a region is $h$ edges from the seed, we store its accumulated cost. This cost provides a baseline value, say $T$; we increase the baseline by a factor $k$, using $h = 10$ and $k = 100$ in our examples. Subsequently, for every pixel we consider adding to the region, we compare the total weight of its *most recent* $h$ edges, say $W_h$, with the baseline $T * k$. When $W_h > T * k$, we terminate progression. In other words, any increase beyond a factor of $k$ in the rolling-average path cost halts region growth in that direction. Algorithm 1 shows the steps.

### 3.3 Rendering

Once we have assigned preliminary region IDs to all pixels, we finalize the IDs by performing a connected component analysis of the ID map, a process we refer to as *flattening*. Doing so gives us different labels for region fragments that have been separated by other regions. For each of the final flattened regions, we compute the mode of its color histogram and apply this color to its constituent pixels. The final result is an oversegmented abstraction of the image. Figure 4 shows region contours resulting from rendering the connected components. Note that our approach is able to convey textured areas of the image, such as the grass and tree trunk, by irregularly-shaped regions. Often, the parts of the image that have little detail do not generate any seeds, and consequently, these areas may not be assigned any region ID before the region growing process completes. The connected component analysis guarantees that all pixels will be given a region ID.

However, uniform areas of the image may be covered by large regions with little interior detail. For many applications, this may be desirable. For cases where it is not, we suggest covering uniform portions of the image by a template of textured regions from another image. In Figure 5, we show an example of a textured image and the corresponding regions, suitable for use as template regions.

---

**Algorithm 1** Region Growth

**Input:** G: A graph on input image , s : seed , k : termination parameter

**Output:** R : Generated region

1: **procedure** GENERATE REGION($G, s$)
2:     Initialize pixel costs to infinity
3:     Set seed cost to $0$
4:     Push the current seed to an empty priority queue $Q_{reg}$
5:
6:     **while** $Q_{reg}$ not empty **do**
7:         Pop the top pixel $u$ and
8:         Set the total cost C to $u$.cost
9:         Push $u$ to $R$
10:         **for all** neighbor v of u **do**
11:             alt := C + $D(s, v)^2$
12:             **if** alt $<$ $v$.cost **then**
13:                 $v$.cost := alt
14:                 **if** $u$ passed $h$ hops for the first time **then**
15:                     $T$ := $v$.cost
16:                 set A to the h'th ancestor of u
17:                 W := v.cost - A.cost
18:                 **if** $| W | < T * k$ **then**
19:                     $v$.parent := $u$
20:                     Push $v$ to $Q_{reg}$
        **return** R

---

## 4    REGION MANIPULATION FOR STYLIZATION

We can use the regions directly for image abstraction. In addition, we can manipulate the regions, for example altering region shapes or manipulating region colors. In this section, we undertake a preliminary discussion of some of the possibilities.

### Contour simplification

Drastic shape simplification has been used for stylization previously [10, 28], often using highly simplified shapes. We simplified the region boundaries based on the Ramer-Douglas-Peucker Algorithm [8], which simplifies a piecewise linear curve by removing vertices based on a measure of error. This method would keep the shape of the structures very close to the original boundary; because

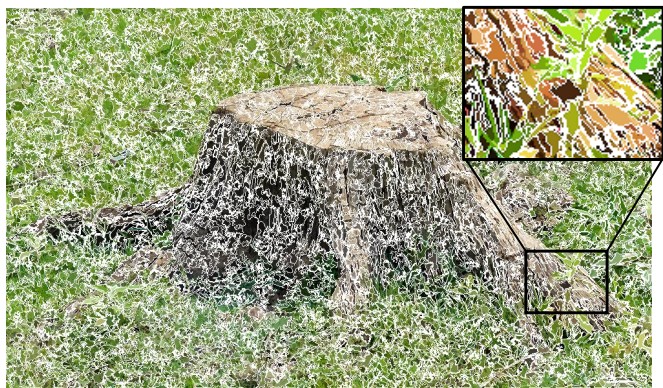

Figure 4: Abstracted stump image. Region boundaries are shown in white.

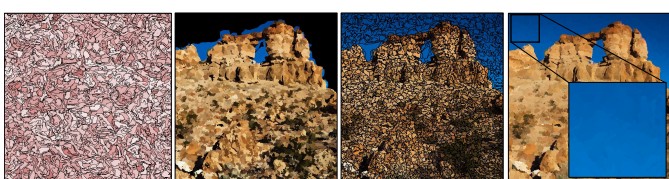

Figure 5: Noise used as background texture for the sky in the arches image.

we began with elaborate shapes, we retain a certain level of complexity even after simplification. A user can decide the level of simplification by adjusting the threshold $\epsilon$ for computing the error. In Figure 6, $\epsilon$ is set to 4 for both images. Higher thresholds create more drastic shape simplifications. We showed the simplified boundaries of initially generated regions and the flattened ones. Simplifying the flattened regions will create gaps between connected regions. We used a smoothed version of the original image to cover the background. Boundary simplification of the larger regions can produce interesting results, though the effect on small regions is negligible. We rendered the simplified boundaries of the original regions in all of our results.

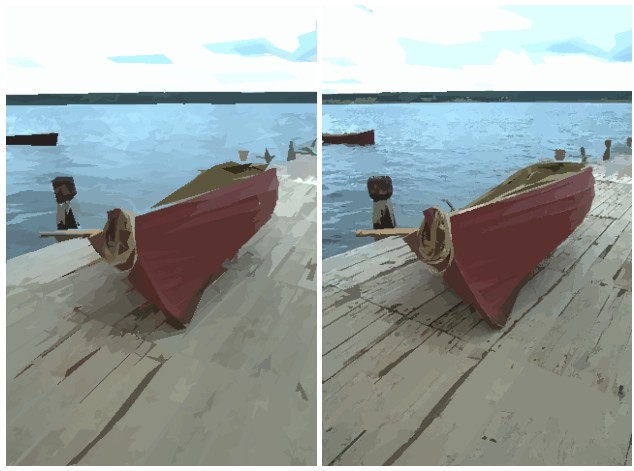

Figure 6: Abstractions after boundary simplification by Ramer-Douglas-Peucker Algorithm ($\epsilon = 4$), with initially generated regions (left) and with flattened regions (right).

**Boundary smoothing**

Alternatively, we suggest smoothing the region boundaries in frequency space, computing Fourier shape descriptors [3, 15] and then truncating the coefficient sequence. This process produces a high level of abstraction. Some results appear in Figure 7, where we retained the first 7 coefficients.

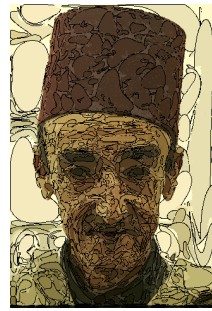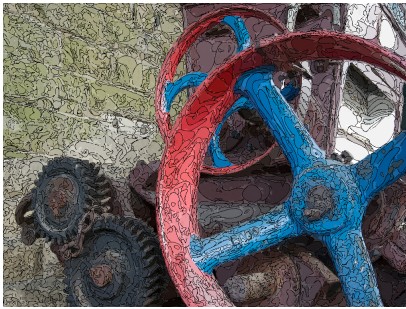

Figure 7: Regions simplified with boundary smoothing with Fourier shape descriptors.

**Reduced color palettes**

We modified the regions' colors, restricting them to small color palettes. The smallest possible color palette is a black and white palette; see Figure 8, where we threshold each region separately. Despite the simplicity of the approach, the oversegmentation lets us capture small details such as the fur in the *cat* image and thin structures in the *port* image.

Choosing a color palette and automatically assigning colors to regions in general is not trivial. Here, we show a simple technique, recoloring the regions with a color palette extracted from an input image. The main colors are determined using k-means; the output color for a particular region is the nearest palette color to the region's average color. A sample result is given in Figure 9. The technique is a generalization of thresholding to an arbitrary palette.

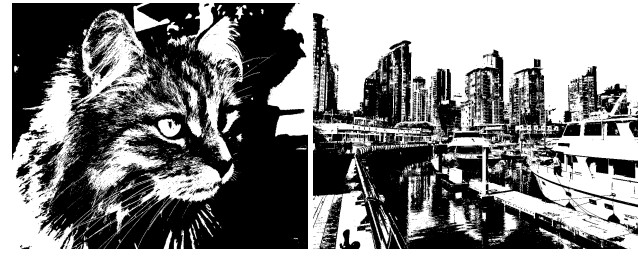

Figure 8: *Cat* and *port* images in black and white.

## 5 RESULTS AND DISCUSSION

We give more examples of our region-based abstraction in Figure 12. Original images for other examples presented in this paper can be seen in Figure 13.

We encourage readers to zoom in to better observe the abstraction effects. We wanted thin features and textures to stay visible after abstraction. In the *market* image, the texture of the bale of hay has been captured, and boundaries of the wheat sheaves kept their jagged shapes. Some small objects on the back shelves and hanging objects are abstracted. Our abstraction preserved strong edges such as the bars of the booth. On the right, the simplified version of the abstraction by means of overlapped regions created more artistic results.

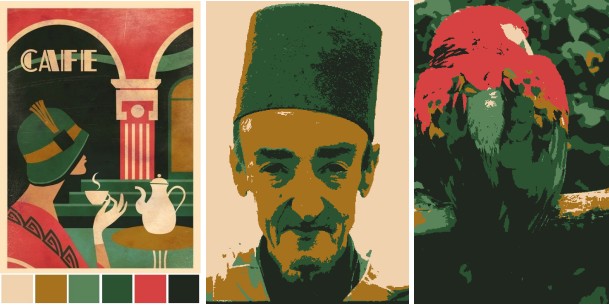

Figure 9: Recoloring using a color palette from Art Deco Poster. Left to right: Art Deco poster and extracted color palette; recolored *old man*; and *parrot*.

The *temple* image shows how our abstraction mechanism can keep both isotropic structures and anisotropic textures at the same time. The abstraction process retained irregular textures in the background. The bricks have low contrast textures which are not prominent but are still captured by the regions. Abstraction through boundary simplification deforms the regular structure of the bricks, which might be undesirable.

Images of nature such as the *autumn* image often contain irregular features like masses of leaves and bark textures. Our abstraction mechanism can keep the complexity of these features. The abstraction with simplified region boundaries gives a more severe stylization that nonetheless conveys the content of the image. Unlike the *temple* image, where directionality of features generated a slight undesirable effect, the less structured textures of *autumn* remain comprehensible.

Our abstraction retained the curvy shapes of the hair layers on the dog's ears and chest in the *dog* image. The background grass is out of focus and the original image is blurred. However, with abstraction, the area became more interesting textures. In the rightmost abstractions, we eliminated small regions, which prevented the eyes of the dog from being rendered. A user-supplied detail mask would fix this problem, though the method would no longer be strictly automated.

In our examples, the regions were colored using original image colors. Modification of colors can generate artistic images as well. Some abstraction methods like artistic thresholding [32] used mean-Shift [5] to segment the image. For our black and white results, we used thresholding of our oversegmentation. Since this naive method does not consider the connectivity of the regions, the effectiveness of the result is based strictly on the complex shapes of the regions we provided.

### 5.1 Comparison with oversegmentation methods

Our priority-based overlapping region creation allows us to preserve small structures. We briefly compare our segmentation with those created by SLIC, Felzenszwalb's method [11] and *QuickShift*. SLIC was specifically intended to create compact, uniform-size segments; QuickShift has more variation in region size and its segments are not necessarily compact, but it still struggles to capture thin structures. The size and number of segments can vary in Felzenszwalb's method, depending on local contrast.

Figure 10 shows oversegmented regions produced by SLIC, *QuickShift*, Felzenszwalb's method, and our method. In general, our regions are far more irregular, a property undesirable for some applications but useful in stylization. Our method is capable of keeping directional patterns. The regions on the water are oriented parallel to the gradient. In the SLIC and *QuickShift* methods, the vertical boundaries cut across the shadows on the water and the curve on the arch of the bridge. Felzenszwalb's segmentation generates regions quite similar qualitatively to ours. For this application, many of the differences are unimportant. Still, Felzenszwalb's regions do not capture well the large structures such as the arch of the bridge. Our segmentation provides more definition in regions with gradients and small details; for example, see the branches and the area beneath the bridge.

### 5.2 Performance

Our algorithm was implemented in C++, using OpenCV for some functionality. On an Intel(R) Core(TM) i7-6700 with a 3.4 GHz CPU and 16.0 GB of RAM, our unoptimized research code takes approximately 67 seconds to process a $1024 \times 680$ input image with about 4200 initial seeds, while the processing time for an image of size $480 \times 320$ with 1200 seeds is about 5 seconds. The majority of the computational expense is incurred by the region growth process.

### 5.3 Limitations

We intended our algorithm to create highly complex regions, making use of local image detail to guide region shapes. However, it works best when there is significant local color variation. When applied to blurred or smooth images, the lack of detail can lead to more uniform region shapes. For example, consider the blurred background in the *toque* image and the smooth surfaces in the *tomatoes* image, shown in Figure 11.

## 6 CONCLUSIONS AND FUTURE WORK

In this paper, we presented an image stylization technique that creates intricate spatial primitives using a region growth mechanism, customized to the target image, then covers the image plane with these primitives. The results strike a balance between detail removal and preservation. Further abstraction is possible through simplifying the region boundaries and recoloring the regions.

There remain several directions for further development. The initial seed placement could be adjusted; it should be possible to obtain equally complex results with fewer seeds, saving some computation time. We would like to experiment with different distance metrics for the initial region calculation, using not only color in the incremental distance but also other image properties, including larger-scale features such as histograms of gradients. Applying the method to video, computing regions in 3D rather than 2D, is a natural extension.

We have obtained appealing abstractions using the cumulative range geodesic, and presented some preliminary results of manipulating these boundaries and coloring the regions. We would like to explore more sophisticated methods for adjusting boundaries and for coloring the region interiors.

#### ACKNOWLEDGMENTS

The authors thank NSERC and Carleton University for financial support. Thanks go to Oliver van Kaick and other members of the GIGL group at Carleton University for valuable discussions about this work. Many images were provided by photographers who made their work available through Flickr. Thanks to the following contributors: Theen Moy (cat), sicknotepix (girl), Greg Myers (tomatoes), James Marvin Phelps (arch), David Taylor (port), Nicholas A. Tonelli (autumn), Mark & Lesley (owl), Carl Mueller (machinery), Carsten Ullrich (bridge), Mongkhon Pookpun (stump), Michal Ščuglík (dog), Rosa Azami (market). Other images (old man, starfish, temple, boat), came from The Berkeley Segmentation Dataset and Benchmark [21].

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
