# OpenReview forum: "Image Abstraction through Overlapping Region Growth"
_graphicsinterface.org/Graphics_Interface/2020/Conference — GI 2020_

### Official Review · AnonReviewer3 · 2020-04-20
**Image stylization that produces irregular super-pixel effect**

**Rating:** 6
**Confidence:** 2

**Review:**

This paper presents an image stylization technique that produces a different super-pixel effect where pixel boundaries are highly irregular and aligned with the feature of the original image. Overall, I like the artistic style and it has different characteristics compared to previous methods.

The paper is well-written and limitations of the method are clearly carved out. Some captions (e.g., Figure 4,5,7,8,10) could be improved to have self-contained explanations.

My major concern is the runtime. It seems that the proposed stylization is mainly served as a design tool. Having runtimes in the order of minutes may hinder practical usage. I would like to see more discussion on the computational bottleneck and how to improve the performance.

Another concern is the motivation of this particular style. It would be nicer to motivate the proposed style with either applications that prefer to have this style or some art pieces that exhibit this style.

I admit that I am not an expert in this field, it is a bit difficult for me to access the technical contributions compared to a plethora of image stylization techniques. But the results are plausible, extensions and limitations are well-discussed, thus I am leaning towards accepting this paper.

---

### Official Review · AnonReviewer2 · 2020-04-20
**Interesting work with insufficient comparisons**

**Rating:** 6
**Confidence:** 3

**Review:**

The seed placement algorithm seems to make sense — is there any rationale behind it? If this is original, please give more details. Otherwise, if it is inspired by some other prior work, please reference them accordingly.

Section 4 discusses quite a few applications. Most of them would be better demonstrated if compared with an existing method. For example, the reduced color palettes to black and white indeed seem impressive. It would be nice to show what a naive method would produce as well as how a strong existing baseline method would perform.

Figure 12 shows several comparisons with [DP73]. Overall I prefer the ones in the middle column. However, regardless of the algorithmic details, as I zoom in, it seems like the level of details is quite different in the [DP73] column and the middle column. For example, there are a lot of details in the Chimney sky and the leaves. Is epsilon of 4 a comparable parameter? If [DP73] does not abstract the image so far, would the result be comparable to the proposed method?


How do the flat regions change with a video or an animated scene? One known issue with video stylization is to keep the regions somewhat stable across frames. I wonder if the proposed method has some insight in terms of temporal consistency.

---

### Official Review · AnonReviewer1 · 2020-04-20

**Rating:** 7
**Confidence:** 3

**Review:**

The paper proposes an "image abstraction" (i.e. simplification of image content for non-photorealistic rendering) algorithm based on region growing.  The approach starts with a seed placement stage where each region is initialized from a set of connected components of a difference image between the original and a blurred version.  Then, a region growing mechanism is used to grow up to a per-region local threshold.  The grown regions are then rendered in order of size to produce the output.

The paper is appropriate to the GI community, is clearly written, and to my knowledge presents an original algorithm for image abstraction.

Strengths:
+ Straightforward algorithm with interpretable parameters should allow for ease of user-direction of the results
+ Qualitative results show good quality relative to other traditional over-segmentation based approaches from prior work.

Weaknesses:
- No quantitative evaluation or user study to validate benefit of proposed algorithm relative to baselines

Despite the lack of quantitative evaluation, I believe the paper proposes a technically sound algorithm which appears to produce good quality image abstractions.  I am therefore in favor of acceptance.

---

### Meta-Review · Area_Chair1 · 2020-04-22

**Recommendation:** Accept
**Confidence:** 4

**Metareview:**

All the reviews are positive about the results demonstrated in the paper. It would be good to address the quantitative evaluation concern, clarify the runtime/bottleneck, and add corresponding comparisons.

---

### Decision · Program_Chairs · 2020-04-25

Accept